# Dissemination of Enterococcal Genetic Lineages: A One Health Perspective

**DOI:** 10.3390/antibiotics12071140

**Published:** 2023-07-01

**Authors:** Joana Monteiro Marques, Mariana Coelho, Andressa Rodrigues Santana, Daniel Pinto, Teresa Semedo-Lemsaddek

**Affiliations:** 1Centre for Interdisciplinary Research in Animal Health (CIISA), Faculty of Veterinary Medicine, University of Lisbon, Av. da Universidade Técnica de Lisboa, 1300-477 Lisbon, Portugal; jmarques@fmv.ulisboa.pt (J.M.M.); fc57492@alunos.ciencias.ulisboa.pt (M.C.); fc57820@alunos.ciencias.ulisboa.pt (A.R.S.); daniel.fs.pinto@tecnico.ulisboa.pt (D.P.); 2Associate Laboratory for Animal and Veterinary Sciences (AL4AnimalS), 1300-477 Lisbon, Portugal

**Keywords:** *Enterococcus*, genetic lineages, virulence factors, antimicrobial resistance, One Health

## Abstract

*Enterococcus* spp. are commensals of the gastrointestinal tracts of humans and animals and colonize a variety of niches such as water, soil, and food. Over the last three decades, enterococci have evolved as opportunistic pathogens, being considered ESKAPE pathogens responsible for hospital-associated infections. Enterococci’s ubiquitous nature, excellent adaptative capacity, and ability to acquire virulence and resistance genes make them excellent sentinel proxies for assessing the presence/spread of pathogenic and virulent clones and hazardous determinants across settings of the human–animal–environment triad, allowing for a more comprehensive analysis of the One Health continuum. This review provides an overview of enterococcal fitness and pathogenic traits; the most common clonal complexes identified in clinical, veterinary, food, and environmental sources; as well as the dissemination of pathogenic genomic traits (virulome, resistome, and mobilome) found in high-risk clones worldwide, across the One Health continuum.

## 1. *Enterococcus* spp.—An Overview

*Enterococcus* spp. are ubiquitous gram-positive cocci that inhabit the normal intestinal microbiota of humans and animals and colonize a variety of environmental sources such as soils, water, and fresh and fermented foods. These facultative anaerobes, catalase-negative, non-spore-forming bacteria can tolerate high salt concentrations (6.5% NaCl), wide pH values (4.5–10.0), and growth temperatures ranging from 10 to 45 °C (optimal at 36 °C) and present variable resistance to chemical disinfectants, such as alcohol and chlorhexidine [1,2,3]. In fact, enterococci possess remarkable stress resilience, which can be enhanced by exposure to sublethal harsh conditions, surviving temperatures as high as 60 °C, desiccation in the environment, and starvation [4]. In addition, enterococci are known to hydrolyze esculin in the presence of 40% bile salts [1,2]. As homofermentative bacteria, they produce lactic acid from glucose fermentation without gas production [2]. Likewise, enterococci play a significant role in the organoleptic properties of fermented foods, due to their natural occurrence as contaminants in raw and dairy food products, resistance to food processing conditions (thermotolerance), low toxicity, and ability to acidify the environment [2]. These attributes are crucial for successful use in food fermentation processes, either as starter or non-starter bacteria, and inclusion in probiotic formulations [5].

However, enterococci have emerged as relevant opportunistic pathogens in the last decades, representing a major cause of healthcare-associated infections due to rapid adaptation to the host and ability to acquire a diversity of virulence and antibiotic resistance genes (ARGs; resistome [6]) through horizontal gene transfer (HGT), largely mediated by mobile genetic elements (MGEs; composing the mobilome) [7]. These bacteria can cause a variety of infections in humans (urinary tract, intra-abdominal, and tissue infections; bacteremia, and endocarditis [8]) and animals (urinary tract infection, otitis externa, peritonitis, endocarditis, and mastitis in dairy animals [9]). Among the growing number of enterococcal species identified, *Enterococcus faecalis* and *E. faecium* are the most common infection-related species, with *E. faecalis* accounting for the majority of infections [10]. Yet, the proportion of *E. faecium* infections has been progressively increasing, due to the spread of antimicrobial resistance (AMR), particularly vancomycin-resistant *E. faecium* (VREfm), which has been responsible for the majority of vancomycin-resistant enterococci (VRE) outbreaks worldwide [11,12]. According to recent evidence, the percentage of clinical VREfm varies between 1 and 46.3% in Europe, 75 and 80% in the USA, and 11.3 and 75% in Australia [11]. These increased levels of resistance have led to the inclusion of VREfm in the high-priority pathogens list, defined by the World Health Organization (WHO) and the ESKAPE pathogen group formed by *E. faecium*, *Staphylococcus aureus*, *Klebsiella pneumoniae*, *Acinetobacter baumannii*, *Pseudomonas aeruginosa*, and *Enterobacter* spp. [13,14]. Other relevant species of the enterococcal genus include *E. avium*, *E. gallinarum*, *E. casseliflavus*, *E. hirae*, *E. mundtii*, and *E. raffinosus*, *E. gallinarum* and *E. casseliflavus* being of special interest, due to their intrinsic resistance to vancomycin [2]. In general, it has been estimated that enterococci represent 6.1–17.5% of the total number of isolates obtained from European patients carrying hospital-acquired infections between 2010 and 2020 [15].

The aim of this literature review is to describe enterococci’s most relevant features, focusing on the overall dissemination of virulence and antimicrobial resistance genetic traits across the One Health sectors. A literature search was mainly conducted on PubMed and ScienceDirect databases (including cross-references). Only the most recent reports (with a major focus on the last five years) addressing the dissemination of genetic determinants between sectors of the One Health triad were selected (further clarified in Section 4).

## 2. Genetic Lineages

*E. faecalis* and *E. faecium* exhibit conserved genomes, although they possess a significant accessory genome (up to 38% in *E. faecium*), which contributes to remarkable genomic plasticity [16]. The evolution of enterococci was predominantly influenced by recombination and proficiency in acquiring novel genes through HGT facilitated by MGEs, such as plasmids, transposons [16,17], genomic islands (GI), and prophages [18]. The genome plasticity of clinical *E. faecium* can be explained by the lack of genome defense mechanisms limiting HGT, such as CRISPR–Cas and restriction-modification systems [16]. Accordingly, an inverse correlation between the presence of the CRISPR–Cas locus and acquired antimicrobial resistance in *E. faecalis* was first demonstrated in 2010 [19]. Recently, the lack of CRISPR–Cas genes was also associated with higher antimicrobial resistance rates and multidrug resistance in *E. faecalis* and *E. faecium* originating from hospital wastewater [20].

The duality between commensals and infection-causing organisms, and the remarkable adaptability to various environmental sources, led to the division of *E. faecium* into two distinct genomic clades: clade A, encompassing hospital-associated lineages (e.g., vancomycin-resistant isolates); and clade B, comprising community-associated enterococci [3,21,22]. However, the discovery of clinical isolates within the clade B lineage led some authors to challenge this classification [23]. Palmer et al. [17] pointed out two evolutionary theories to explain the emergence of the two clades. The first is that clade A and clade B may be endogenous to the gastrointestinal tracts of different hosts and now coexist among human microbiota, as a result of antibiotic elimination of competitors. The other is that clades A and B may be diverging from each other, due to antibiotic use and ecological isolation, although this is less likely [17]. Clade A can be further classified into two distinct sub-clades: clade A1, consisting of human clinical epidemic strains, which account for the majority of human infections (mostly referred to as CC17, further explored in Section 4); and clade A2, typically associated with livestock/animal strains and some occasional human infection isolates [22]. Interestingly, some animal-origin enterococci found in clade A1 were isolated from pet dogs, suggesting an interconnection between hospital strains and household pets [22]. In fact, an additional study demonstrated that sub-clade A1 emerged from A2, and that hospital isolates identified as A2 represent a genetic continuum between A1 and community *E. faecium* placed in clade B [7]. Thus, the existence of this sub-clade division has been contested and remains uncertain [7,23].

In comparison to *E. faecium*, *E. faecalis* isolates do not exhibit a multiclade structure, instead displaying a significant reliance on the acquisition of mobile elements as a primary driver of genomic diversity [17]. Nevertheless, some major high-risk enterococcal clonal complexes (HiRECCs), namely CC2, CC9, and CC87 (further explored in Section 4), have been associated to hospital settings and healthcare-associated infections [1,24].

## 3. Enterococcal Fitness Factors, Virulence Traits, and Antimicrobial Resistance

### 3.1. Fitness Factors

Enterococci express various fitness and adaptive factors, referring to cumulative evolutionary mechanisms that allow them to survive and thrive in high-pressure selective environments (e.g., healthcare facilities and food industries). Their durability, and impressive multistress tolerance, optimizes enterococci as competitors of the host gastrointestinal tract and allow them to persist in healthcare environments, despite careful disinfection procedures, and they acquire antimicrobial resistance genes from other bacteria present in the same niche [25]. A key feature contributing to the enhancement of enterococcal fitness is the ability to produce biofilms (partly regulated by quorum-sensing mechanisms), leading to rapid adhesion onto abiotic surfaces, tissue colonization, and higher stress tolerance [26]. Some examples of molecular mechanisms involved in stress tolerance regulation include oxidative stress (in some *E. faecalis* catalase-positive *katA*; peroxidases; and *sodA* superoxide dismutase), thermal stress (heat stress regulated by heat-shock proteins; and cold stress by physiological changes, expression of cold shock proteins, and cold acclimation proteins), pH stress (alkaline mediated by Na^+^ (V_1_V_0_)-type ATPase and K^+^/H^+^ antiporter in *E. hirae*; acid mediated by F_1_F_0_-APTase), osmotic stress (ion transport systems, chaperones, and alterations in the physiochemical properties of the cell envelope), metal stress (copper, manganese, and iron), nutritional stress (nutrient-sensing global regulators CcpA, CodY, and (p)ppGpp messenger), antibiotic stress (associated mechanisms further explored in Section 3.3), host-related stresses (antimicrobial peptides, bile, and blood), disinfectants, among others (reviewed by Gaca and Lemos [25]). Overall, stress tolerance and pathogenesis are closely interlinked, considering that both contribute to enterococcal resilience and survival in harsh environments [25].

### 3.2. Virulence Traits

Virulence factors are bacterial traits that result in host infection [3]. These determinants are encoded by virulence genes, some of which can be found in pathogenesis islands (PAIs) or plasmids, thus being important to consider in the dissemination of specific genetic lineages across the One Health continuum. Previous reports have shown that clinical enterococcal isolates present the highest virulence, with *E. faecalis* having the leading role [1], followed by food isolates and starter strains [27]. Virulence factors can be divided into two main groups: (a) externally secreted factors, e.g., cytolysin, gelatinase, secreted antigen A, serine protease, and glycosyl hydrolase; and (b) cell-surface factors, e.g., extracellular surface proteins (Esp), aggregation substances, adhesins, pili, and others [1,5]. 

Cytolysin is responsible for a bactericidal effect against other gram-positive bacteria and hemolytic disruption of blood erythrocytes [8,28]. It is commonly encoded on plasmids [29], also being found on the chromosome [3], and is regulated by a quorum-sensing mechanism [27]. Gelatinase is a member of the metalloprotease extracellular zinc-endopeptidase family, able to hydrolyze gelatin, collagen, casein, and other peptides [30]. Encoded by the *gelE* gene, located on the chromosome, it is co-transcribed with the serine protease SprE, by the FSR system [1,8]. Both cytolysin and gelatinase have been mostly studied for *E. faecalis* [30]. Secreted antigen A (SagA) is a protein exclusively present in *E. faecium*, with a role in stress activation and cell growth, being putatively involved in cell wall metabolism [31]. In addition, SagA containing a repeated region in clade A1 enterococci appears to improve biofilm formation, in contrast to SagA present in strains from clade B [32]. Glycosyl hydrolase, mistakenly named hyaluronidase in *E. faecium*, is a degradative enzyme encoded by the chromosomal *hyl* gene, acting on hyaluronic acid present in cellular tissue and facilitating the spread of enterococci and secreted toxins through the host tissues [27,30]. However, there is still no proof that it contributes to enterococcal colonization and pathogenesis [33]. 

Regarding cell-surface virulence factors, these are important for bacterial defense mechanisms, such as biofilm formation and immune system evasion [1]. The enterococcal surface protein (Esp), mostly present in *E. faecalis* and *E. faecium* clinical isolates, is an adhesin associated with colonization and persistence in urinary tract infections [8] and has an important role in biofilm formation and antimicrobial resistance [34,35]. The *esp* gene can be encoded in PAIs and may be transmitted by conjugation [5]. Aggregation substances (known as AS or Agg) are pheromone-inducible surface proteins, responsible for the adhesion to both prokaryotic and eukaryotic cells [5,8], also aiding in colonization. Aggregation substances are mediated by plasmids—pPD1 contains genes encoding the Asp1 protein, pCF10 contains genes encoding the Asc10 protein, and pAD1 contains genes encoding the Asa1 protein [5]. Microbial surface components recognizing adhesive matrix molecules (MSCRAMMs) are a subfamily of adhesins that are able to bind to extracellular matrix elements. Three types of MSCRAMMs have been described so far in *Enterococcus*: (1) Ace (adhesion of collagen from *E. faecalis*), which plays a role in endocarditis pathogenesis together with (2) Acm (adhesion of collagen from *E. faecium*), and, lastly, (3) Scm (second collagen from *E. faecium*) [1,27]. Pili are composed of multimeric fibers made of pilin subunits. In *E. faecalis*, pili are encoded by two distinct gene clusters: *ebp* (endocarditis- and biofilm-associated pili) and *bee* (biofilm formation) [1].

Other factors accounting for enterococcal virulence are PrpA, a thermosensitive protein that binds fibrinogen, fibronectin, and platelets; and phosphotransferase system (PTS) transmembrane proteins and PTS-associated BepA, a permease that contributes to biofilm formation in human serum, being mostly present in clade A isolates (reviewed by Gao et al. [30]).

A scheme of the most important virulence factors described in *Enterococcus* spp. is displayed in Figure 1, some of which are explored further in Section 4.

### 3.3. Antimicrobial Resistance and Associated Mechanisms

AMR refers to the ability of bacterial cells to withstand and avoid the harmful effects of antimicrobial agents [36], leading to difficulties in treating infections caused by resistant bacteria. As opportunistic pathogens, enterococci initially caused infections in immunocompromised patients, and later gained relevance as a major pathogen in hospitals with the introduction of third-generation cephalosporins (β-lactams class), to which these bacteria are naturally resistant [3]. As resistance to ampicillin and ciprofloxacin emerged a decade later [3,5], the glycopeptide vancomycin started being used as an alternative therapeutic option [3]. Moreover, the incorrect and prophylactic overuse of antimicrobials, both in healthcare environments and production animals, led to a progressive increase of resistance among enterococci, even to last-resort antimicrobials [5]. Thus, there is an urgent need for active biosurveillance of the dissemination of ARGs between bacteria present in each sector of the One Health triad.

Briefly, enterococci are known to be intrinsically resistant to a variety of antimicrobials, including cephalosporins, trimethoprim-sulfamethoxazole, and present low-level resistance to β-lactams and aminoglycosides [27]. Healthcare-associated strains belonging to clade A are generally resistant to ampicillin [37] and quinolones [24]. In addition, the animal-associated sub-clade A2, and the community-associated clade B, rarely lead to invasive infection or present vancomycin resistance [22,23]. Multidrug resistance to macrolides, tetracyclines, streptogramins, and glycopeptides has been observed in both clinical and animal/veterinary enterococci [38]. A summary of the main antimicrobial classes, mechanisms of action in enterococcal species, and genes responsible for antimicrobial resistance is presented in Table 1.

VRE, particularly VREfm, has become a growing concern in modern medicine, especially considering that vancomycin is a last-resort antimicrobial to treat hospital-acquired life-threatening infections (HAIs). Accordingly, VREfm, defined as an ESKAPE pathogen, is identified by the WHO as a high-priority pathogen, emphasizing the urgent need for the development of novel antimicrobial treatment options [40]. Glycopeptide resistance (e.g., vancomycin and teicoplanin) occurs through the substitution of N-terminal D-Alanine-D-Alanine in the peptidoglycan synthesis pathway, to either D-Alanine-D-Lactate (D-Ala-D-Lac) or D-Alanine-D-Serine (D-Ala-D-Ser) [41], leading to a lower binding affinity of glycopeptides in the cell. To date, various gene clusters responsible for glycopeptide resistance have been described in enterococci, named *vanA* to *vanN* [42]. Vancomycin resistance is more common in the species *E. faecium*, *vanA* and *vanB* operons being the most frequently detected [1,42,43]. *vanA* is plasmid-located and mediated by Tn*1576* transposon, being characterized by a high degree of acquired inducible resistance to both vancomycin and teicoplanin [5,42,44]. While predominantly observed in *E. faecium* strains, it has also been detected in *E. faecalis* and, to a lesser extent, in *E. durans*, *E. raffinosus*, *E. hirae*, *E. avium*, and *E. gallinarum* [5]. In contrast, the *vanB* resistance phenotype, encoded on the chromosome and carried by Tn*1549*-Tn*5382*, is characterized by variable levels of vancomycin resistance and susceptibility to teicoplanin [42,44]. 

A recently described variant of VRE, vancomycin-variable enterococci (VVE), are susceptible to vancomycin while carrying the *vanA* genotype (VVE-S), with the ability of becoming resistant to vancomycin upon exposure to glycopeptides [42,45]. Several outbreaks of VVE have been reported around the globe [42,45,46,47]. 

Linezolid (oxazolidinone) has been chosen as a last-resource antimicrobial to fight multidrug-resistant and VRE infections, as it presents bacteriostatic activity against a broad range of gram-positive bacteria. However, resistance to linezolid was first observed only 1 year after its introduction (in 2000) and has been increasing among clinical isolates, leading to the introduction of new antimicrobials with variable availability around the world [5,48]. Tigecycline (glycylcycline) and daptomycin (cyclic anionic lipopeptide) are additional last-resource options [42].

## 4. Enterococcal Genetic Traits: Dissemination in the One Health Continuum

According to the WHO [49], around 75% of emergent human infectious diseases originate in animals, due to the interdependence between humans and animals, changes in ecosystems, intensification of agriculture, urbanization, travel, and commerce [50]. To tackle this increasing issue, the “One Health” concept was coined over two decades ago, aiming to define an integrated, multisectoral, and sustainably balanced health approach that recognizes the interconnection between human, animal, and environmental health [51]. As a result of enterococcal duality and transversality to different ecosystems, rapid adaptation to the host, and propensity to acquire virulence and resistance traits, enterococci are fit to be used as indicator organisms for tracking and monitoring the transmission of pathogenicity-related genetic determinants in a One Health perspective [52]. Accordingly, antimicrobial resistance and the dissemination of resistance genes through interconnected ecosystems have been considered quintessential One Health issues [53]. To face this problem, many authors [18,36,38,54,55] have been performing comparative genomic analyses of *Enterococcus* spp. as indicator bacteria, to study the transference of ARGs, virulence genes, and MGE determinants across the One Health continuum. 

Accordingly, considering the extensive number of studies addressing the dissemination of genetic determinants within each sector (human health, animal health, environmental health, and food safety), the present review will focus on the dissemination of genetic traits across sectors of the One Health triad. However, considering that HiRECCs are associated with hospital settings and healthcare-associated infection, human-related studies are discussed in more detail, to assure understanding of the topic under review.

### 4.1. Human Healthcare-Associated Genetic Traits

As aforementioned, enterococci have emerged as one of the most frequent causes of infections in clinical and healthcare settings [56]. Among all species of *Enterococcus* spp., *E. faecalis* and *E. faecium* are the main healthcare-related pathogens, despite a fast-epidemic growth of multidrug-resistant *E. faecium* clones, which has resulted in increased prevalence of *E. faecium* in comparison to *E. faecalis* [15,57]. According to previous studies [29,58] using MLST, there are host-specific genetic clusters of *E. faecalis* and *E. faecium* isolated from hospital-acquired infections, defined as clonal complexes (CCs), in which a variety of sequence types (STs) are assembled. 

From an evolutionary perspective, the gain and loss of virulence and ARGs via MGEs directly influence the emergence of new specific lineages in a particular geographic area, which could quickly disperse in healthcare-associated settings worldwide [59,60]. The existence of certain resistance genes and virulence factors in a specific local environment induces the acquisition of these traits by epidemic clones via HGT, which may influence evolutionary development toward genetic subpopulations, completely adapted to healthcare settings [58].

Table 2 summarizes some genetic features of the most relevant enterococcal clinical clonal lineages. Given the worldwide heterogenous epidemiology, data are displayed focusing on STs.

Most *E. faecalis* isolated from healthcare environments have been associated with HiRECCs such as CC2 and CC9, which have spread globally [29,64,72]. However, in Europe, CC87 may replace CC9 as a potential high-risk complex [63]. Particularly, CC2, CC8, and CC9 lineages have revealed higher acquired antimicrobial resistance; CC2, CC16, and CC87 were associated with multidrug resistance, of which CC2 and CC87 were associated with healthcare-associated infections (HAIs) [73]. 

*E. faecalis* CC2 comprises various strains in which the genetic determinants are variably present, differences being associated with distinct lineages. For instance, CC2 is characterized by the presence of the *aac (6′)-aph(2″)* gene for high-level gentamicin resistance, encoded by *Tn5281* [62,74], *vanB*-type resistance or vancomycin-susceptibility, high-level resistance to β-lactams due to β-lactamase production (conferred by the *blaZ* gene), and resistance to erythromycin by the occurrence of the *ermB* gene [59,61,75]. Additionally, some studies [62,76,77] in Asia detected at least two virulence genes in CC2 and CC87, including *esp*, *gelE*, *cylA*, *asaI*, *ace*, and *hyl*. Although *E. faecalis* CC87 from Poland is not distributed worldwide, as is also the case for CC2 or CC9, these three complexes share several virulence and antibiotic resistance determinants, as illustrated in Table 2. However, a remarkable characteristic of CC87 is the exclusive dissemination of certain MGEs, such as repUS1pVEF1 plasmids, which carry *vanA* resistance in Polish hospitals [78].

Compared with CC2, the other globally disseminated *E. faecalis* healthcare-associated complex, CC9, has not been so well described in the literature, in terms of phenotypic and genotypic features. However, virulence and antimicrobial resistance traits, in common with CC2, have been identified [29,61,64]. 

Furthermore, in *E. faecalis* a wide range of plasmids and transposons are involved in the genetic transmission of resistance and virulence, which may promote the prevalence and dissemination of CC2 and CC9 in healthcare settings [29,79]. Nonetheless, these healthcare-associated clones are not only disseminated in hospitals, but also in the community and in animals, especially CC2 [56]. 

Regarding *E. faecium*, some specific clones have rapidly developed multiple AMR since the 1990s, which allowed adaptation and spread in healthcare environments [65]. Clinical-derived *E. faecium* forms a polyclonal subpopulation of different CCs assembled in clade A, which have separately developed from several ancestral clones, namely MLST sequence type 17 (ST17), ST18, ST78, ST192, and ST203 [17,37,80].

*E. faecium* CC17 or clade A1 have been found in at least five continents, carrying ampicillin, vancomycin, high-level quinolone resistance, and the presence of an accessory genome that contains putative virulence genes, namely *esp* and *hyl* genes [60,68,81,82]. Moreover, *fms* genes are also present in CC17 isolates (e.g., *fms8*, *fms9*, and *fms21*), coding for virulence factors such as MSCRAMM-like proteins [69]. For instance, *fms8* or *acm* seem to contribute to the success of healthcare-associated infections, being related to a higher prevalence of CC17 in hospitals [59].

Recent studies [83,84] have demonstrated significant genome flexibility and specific clonal outbreaks in certain hospitals, resulting in new clonal lineages with novel adaptative genetic traits, belonging to CC17. In Germany hospitals, *E. faecium* ST117 significantly increased from 2010 to 2019 and became a highly widespread lineage of CC17 [85]. This sequence type is frequently linked to *vanB*-type resistance, as seen by the formerly dominant ST192 [85]. In contrast, one of the most prevalent CC17 clones in hospitals in China and Japan has been ST78 VRE, which carries a linear plasmid with a distinctive structural *Tn1546* including *vanA* [86,87,88]. 

As in other countries, the emergence of *E. faecium* clones (belonging to CC17) has also occurred in Australia, namely ST203, ST796, ST1421, and ST17 [67,70,71,89]. Initially, ST203 isolates were susceptible to vancomycin, but the acquisition of *vanB*, encoded by *Tn1549*, was responsible for the emergence of VRE ST203 strains [90]. In 2011, ST203 became the most prevalent cause of *E. faecium* bloodstream infections in Australian hospitals [70]. Three years later, in 2014, ST203 was completely replaced by *vanB* ST796 *E. faecium* [70]. This clone caused several outbreaks in Australia and New Zealand, before quickly spreading to other countries in Europe, such as Switzerland in 2017 [91,92]. The presence of *vanA* in some ST796 clones from Switzerland has also been reported [92]. However, a change in the molecular epidemiology of VREfm occurred between 2013 and 2020 in Australia, leading to a large rise in the incidence of *vanA* VREfm in 2016, due to the emergence of *vanA* ST1421 [71,89]. More recently, a decrease in *vanA* VREfm was observed, mainly due to a decrease in ST1421 and ST1424 clones and an increase in *vanB* ST17, which became the most predominant clone in 2020 [89].

Therefore, some clones are endemic to specific geographic regions. For example, *E. faecalis* CC87 is still present in hospitals in Poland [78]; *E. faecium* ST796 was present in Australia and New Zealand in 2014, being later detected in Europe (Switzerland) [70,93]; the first *E. faecium* ST192 and more recently ST117 in Germany [85]; *E. faecalis* ST525 in Brazil [94]; *E. faecium* ST80 associated with the spread of a *vanA*-containing plasmid and acquisition of a heterogeneous accessory genome in Denmark [95]; and *E. faecium* ST1421, which became the predominant *vanA*-type clone from 2015 to 2019 in some Denmark regions [96], among others.

Overall, the prevalence and spread of enterococcal genetic traits are not just a concern in clinical practice, but also a threat to public health, concerning animals, the environment, and the community, due to enterococcal ubiquity across all sectors of the One Health continuum. 

### 4.2. Dissemination of Genetic Traits across One Health Sectors

Many authors [18,36,38,97] have been focusing on the interplay between human, animal, community, and environmental enterococcal clones, from a One Health perspective. Nevertheless, to the best of our knowledge, no review has provided a roadmap of the clonal spread from human clinical settings to animals, food, the community, and the environment, highlighting the dissemination not only between sectors but also around the globe, as hereby presented.

Table 3 summarizes the distribution of specific genetic lineages, virulence features, and antimicrobial resistance of isolates present across the One Health continuum. Only studies comprising MLST analysis with discriminated STs, respective ARGs, and virulence genes were included. To facilitate interpretation and better highlight the dissemination within and between One Health sectors, STs were grouped within the respective CCs.

Several studies [9,115] performed over the years have demonstrated that specific enterococcal strains or genetic traits are shared between animals (or animal-derived food) and humans. In fact, enterococci are known to be present in the gastrointestinal tract of most animals, and there is abundant evidence that drug-resistant strains can occur in companion, farm, or wild animals and be transmitted to humans, causing zoonotic infections [9,115]. Moreover, the overuse of antimicrobials in both human and veterinary medicine has led to the rapid emergence and dissemination of resistant bacteria, which nowadays are resistant to most antimicrobials used in clinical practice [116]. Furthermore, the usage of several antimicrobial classes is shared between human and animal clinical practice, resulting in an overlap of resistant genes detected in both settings, thus compromising human and animal welfare [117,118].

A recent report [119] observed identical antibiotic resistance to ampicillin, virulence patterns, and bacteriocin production within clusters of clinical, farm animal, and retail meat samples, which indicates the transference of enterococcal genetic traits between animals and animal-based foods. This transmission between humans and animals was also confirmed in a previous report [120], which evaluated the spread of antimicrobial-resistant enterococci between dogs and owners in three different households. The authors [120] identified *E. faecalis* in all three settings with very similar resistance patterns and detected clonal spread between pets and the respective owners within each household, the animals’ body parts, and shared domestic objects. Another study described a 100-fold higher proportion of enterococci, including multidrug-resistant (MDR) *E. faecalis* and/or *E. faecium*, in the fecal microbiota of dogs submitted to antimicrobial therapy after leaving an intensive care unit, in comparison to healthy canines [121]. In fact, dogs were proven to harbor *E. faecium* belonging to STs related to human clinical isolates or hospital outbreaks, for instance ST17 [121,122]. Moreover, enterococcal contamination in medical-associated surfaces at small veterinary hospitals was described by KuKanich et al. [123], who also found MDR *E. faecium* isolates. Regarding studies which applied MLST, a previous report [121] examined enterococcal antimicrobial resistance in a veterinary clinic and demonstrated that dogs from intensive care submitted to antibiotherapy harbored MDR-*Enterococcus*, namely belonging to the species *E. faecalis* and *E. faecium*. In addition, dog food commercially available in Portugal was reported as a reservoir of antimicrobial resistance enterococci with clinical relevance [110]. All the above highlight the importance of habitat sharing as a risk factor for the spread of pathogenic microorganisms and associated genetic determinants. Accordingly, a study by Jernberg et al. [124], focusing on human and mouse microbiota, concluded that resistance due to antimicrobial treatment may be long-lasting, meaning that resistant strains may remain for long periods after the treatment is finished. Hence, this over-time colonization may increase the risk of the zoonotic spread of multi-resistant bacterial strains [125,126,127].

Previous reports [106] have already demonstrated the dissemination of specific genetic lineages between animals and humans. For example, *E. faecalis* ST6 (CC2) carrying pheromone-like *vanA* plasmids is disseminated among human and swine hosts in Europe. Furthermore, Neumann et al. [128] performed a core genome MLST analysis of *E. faecalis* strains collected from human infection and colonization, showing a close relationship with isolates from clinical sewage/wastewater, belonging to ST6. From the 97 ST6 enterococci, 57 were allocated in 34 distinct CTs, indicating high potential for rapid adaptation. The authors [128] also found that human–clinical associated lineage ST6 included most of the VRE observed. Barros et al. [129] identified enterococci belonging to ST6 in natural gilthead seabream recovered off the coast of Portugal. Regarding *E. faecium*, in Portugal, ST132 (CC17) conveying *vanA* plasmids was found to be spread between humans and swine. The ST273 described in the previous study was also found in a wastewater treatment plant in Czech Republic (Table 3) [112,129].

A major study [101], using MLST, investigated the prevalence of antimicrobial-resistant and virulence genes in *E. faecalis* from ducks at slaughterhouses, and the dominant sequence types were ST170, ST593, ST314, ST903, and ST192. The other STs present were represented by a single isolate, and the overall results revealed the presence of 16 distinct CCs. It should also be noted that ST170, a single-locus variant (SLV) of ST82, has already been identified in a hospitalized patient in Poland. Additionally, ST116 recovered from duck skin is an SLV of ST145, which has previously been isolated in hospitalized patients from Spain. CC93 comprises four STs, including ST75 (already found in a hospitalized patient from Portugal), ST657 (in a hospitalized patient from China), ST226 (from poultry meat), and ST98 [101]. Overall, the abovementioned data highlight that *E. faecalis* isolated from ducks are associated with human healthcare-related enterococci, which may pose a serious risk to public health. Furthermore, regarding antimicrobial patterns, the vast majority of the duck isolates were resistant to tetracycline, doxycycline, erythromycin, and norfloxacin, which may be associated with therapeutic failure. Other noteworthy STs include ST29, which was identified in poultry, wild birds, rodents, and non-hospitalized humans [130] and in pigs from six different Brazilian states [131]; ST13, which was recovered from poultry in Sweden [132]; ST242, ST214, and ST955, which were all found in bovine feces [104], with ST242 being also discovered in wild Magellanic Penguins [133]; ST214, which was observed in non-hospitalized humans [134]; and ST955, which was found in a hospitalized human in Canada [135]. Other studies, such as the one performed by Fatoba et al. [105], demonstrated the presence of the clinical *E. faecium* CC17 and transmission of multiple STs carrying ARGs and MGEs, from poultry litter to agricultural soil. 

In Malaysia, a research group analyzed the genetic variability and evolutionary relationships of VRE isolated from humans, poultry, and swine [108]. Regarding VREfm, enterococci belonging to ST203 (CC17) were found in both humans and poultry. This sequence type has been associated with healthy individuals, farms, slaughterhouses, and the environment. Moreover, another isolate of ST17 (CC17), one of the most widespread genetic lineages associated with human infections, was identified in swine- [136] and poultry-based products [111].

Overall, antibiotic-resistant bacteria dissemination through food-producing animals or the food chain is considered a public health problem, but its real effect as an impact on human health continues to be uncertain [137]. The rise of AMR poses significant challenges to global health and food safety, to the extent that the WHO anticipates entry into the post-antimicrobial era during the present century [138]. Numerous studies [139,140,141] addressed antimicrobial resistance patterns and virulence factors in food products. Regarding the dissemination of ARGs and virulence determinants, some authors [111,119] have found evidence of clinical antimicrobial-resistant *Enterococcus* disseminated across the food chain. Indeed, meat was indicated as a transmission pathway of *E. faecium* and ARGs from animals to humans. Gouliouris et al. [142] performed a genomic survey of the presence and ARG relatedness of VREfm isolates from livestock farms, retail meat, WWTPs, and bloodstream infections in the United Kingdom and Ireland. Although the authors [120] observed a decrease in VREfm prevalence (between 2003 and 2018), VREfm was isolated in retail meat and was considered ubiquitous in WWTPs. Furthermore, the majority of human- and livestock-related isolates were genetically distinct, and limited sharing of acquired ARGs was observed between healthcare-associated enterococci and livestock. However, Freitas et al. [119] demonstrated that human clinical ampicillin- and multidrug-resistant *E. faecium* clones disseminated in Tunisia were not only genetically identical to clinical enterococci found in other countries and continents, but also identical to strains isolated from animal sources and food in this country. 

Regarding the spread of enterococcal-related resistance and virulence within food samples, López et al. [111] demonstrated that *E. faecium* belonging to CC17 is present in poultry, veal, and rabbit. More recently, Elghaieb et al. [143] investigated the presence of *poxtA-* and *optrA*-carrying linezolid-resistant *E. faecalis* in food-producing animals and retail meat in Tunisia. OptrA is a gene responsible for acquired linezolid resistance and has been rapidly spreading among enterococci of various origins. The authors [143] also highlighted that enterococci recovered from food-producing animals and retail meat in Tunisia carried a plasmid fragment similar to the one found in enterococcal human and swine samples from China. Additionally, some *E. faecium* from animals and *E. faecalis* from the community displayed genetic profiles related with healthcare-associated epidemic strains belonging to clade A1, specifically ST17, ST18, and ST78. However, the community’s *E. faecalis* was mostly associated with healthcare sequence types, such as ST6, ST40, and ST116. Supporting the results by Elghaieb et al. [143], Ni et al. [144] recently isolated enterococci strains from pig carcasses and environmental samples in duck slaughterhouses in China. Approximately 11% of the isolates were resistant to linezolid, with a portion being *poxtA*-positive (1.7%) and *optrA*-positive (16.2%), demonstrating the widespread existence of oxazolidone resistance genes through the food chain to humans [144]. The detection of the same linezolid-resistant enterococci in both food-producing animals and retail meat suggests a potential role of the food chain in the dissemination of this trait, even without selective pressure. Furthermore, the same study also identified the presence of virulence genes typically found in clinical *E. faecium* from clade A1, including *sgrA*, *ecbA*, *esp*, and *acm*. These findings indicate the putative pathogenicity of those enterococcal isolates and highlight the possibility of transmission to humans through food [143]. Additionally, another research group performed a molecular characterization of high-level gentamicin-resistant (HLGR) *E. faecalis* harboring aminoglycoside-modifying enzymes and virulence genes in retail poultry meats from Korea [100]. The authors [100] described ST82 as a foodborne HLGR *E. faecalis* clone, with a track record of outbreaks in poultry in other countries. 

Although enterococci can persist and survive in both fresh and marine water, there is still a lack of consensus regarding their ability to grow and multiply in these environments, due to the scarcity of nutrients [2,145]. Despite this fact, enterococci can be isolated from a panoply of environmental settings with variable availability of nutrients, such as soil, freshwater, beaches [146], and wastewater treatment plants (WWTPs) [147,148]. Specifically, the latter are considered a cost-effective point of control for the dissemination of AMR determinants in the environment, as WWTPs gather excreted urine and feces from urban surrounding areas [102]. Accordingly, *E. faecalis* and *E. faecium* are used as indicators of fecal contamination in water [102]. A comparative genomics study [102] of VRE and MDR enterococci isolated from WWTPs recently performed, described a variety of *E. faecium* and *E. faecalis* STs and associated ARGs, virulence genes, and MGEs (Table 3), pointing to WWTPs as key to achieving a better understanding of the linkage between enterococci present in this setting and AMR dissemination [102]. Another recent study [148] detected VREfm and ampicillin-resistant *E. faecium* hospital-adapted lineages in municipal WWTPs, whose levels in untreated wastewater were significantly higher than directly received hospital sewage. From 28 different ARGs found in the hospital-adapted *E. faecium* clade, 23 were present in bloodstream infection, hospital sewage, and municipal WWTP enterococci, indicating a genetic overlap between isolates from all sectors and a potential ARG transmission route [148]. An additional study [149] detected *vanA*-positive *E. faecium* ST17 and ST80 (CC17) in hospital wastewater, once again highlighting the possible dissemination of healthcare-associated antimicrobial-resistant strains to the environment and community through wastewater. For instance, the ST78 strain from the vast CC17 was associated with isolates derived from WWTP samples, according to [107]. Another study [112] identified the same genetic lineage in WWTP from the Czech Republic. Those ST78 clones were also isolated from bovine meat/milk in Tunisia samples [143,150], which described this complex clone as one of the most common in Chinese cities, leading to its spread throughout the world and indicating clonal persistence in different settings. Sadowy and Luczkiewicz [107] described a set of distinct STs (e.g., ST66, ST361, and ST386) (Table 3) in the Gulf of Gdansk, Poland in wastewater and marine outfalls, influencing the quality of waters. ST66 was also found in water sources for agriculture/recreational purposes in Australia [151]. One study [103], which focused on VREfm ST133 isolated from an aquatic environment in Switzerland (Table 3), demonstrated a close phylogenetic relationship to ST133 recovered from swine feces, highlighting the role of surface water in VRE dissemination. 

Sanderson et al. [18] explored the mobilome and resistome of *E. faecium* across two disparate geographic locations (Canada and United Kingdom) and found that the dissemination of *E. faecium* MGEs and antimicrobial resistance determinants is not limited by within-species phylogeny, habitat (clinical, agriculture, municipal and agricultural wastewater, and natural water sources), or geography. Nevertheless, the authors [18] found association between specific gene profiles and MGEs with the habitat and pointed out the possibility of enterococcal pathogenicity features being associated with a group of clinical/municipal wastewater genomes, which might constitute a human-associated ecotype within clade A. Recently, a One Health comparative genomic analysis of virulome, resistome, and mobilome in *E. faecium* and *E. faecalis* isolated from livestock, human clinical samples, municipal wastewater, and other environmental sources was performed [36]. The study [36] identified 31 *E. faecium* and 34 *E. faecalis* ARGs (62 and 68% being plasmid-associated, respectively), with tetracycline (*tetL* and *tetM*, frequently found on pM7M2) and macrolide (*ermB*) resistance being described in all sectors of the One Health continuum. Additionally, Noh et al. [152] also observed the presence of *tetM*, *ermA*, and *ermB* genes in virulent-MDR *E. faecalis* strains recovered from poultry in Korea, highlighting the spread of tetracycline and macrolide resistance genes among enterococcal isolates from various sectors.

## 5. Conclusions

In general, it is fundamental to emphasize that a diversity of enterococcal genetic lineages has been observed amongst animals, food, community, and human patients. The widespread nature of enterococci-carrying emerging ARGs and virulence genes between and within humans, animals, and the environment is known to occur, regardless of antimicrobial pressure. The present review aimed to highlight the need for effective genomic biosurveillance studies, focusing on enterococci-carrying virulence and/or antimicrobial resistance genetic elements, across sectors of the One Health continuum, and to expose the urgent need for global measures addressing the spread of MDR and highly pathogenic variants. It is imperative to address the issue of patients acquiring difficult-to-treat healthcare-associated infections upon visits or admission to healthcare facilities. However, it is also essential to acknowledge the potential for those microorganisms to be acquired from community-related settings, including the food chain and contact with domestic animals. Overall, the selective environmental pressure granted by the incorrect use and overuse of antimicrobials in humans, agriculture, pets, and livestock further intensifies the exchange of genetic elements between the One Health sectors, as represented in Figure 2. 

## Figures and Tables

**Figure 1 antibiotics-12-01140-f001:**
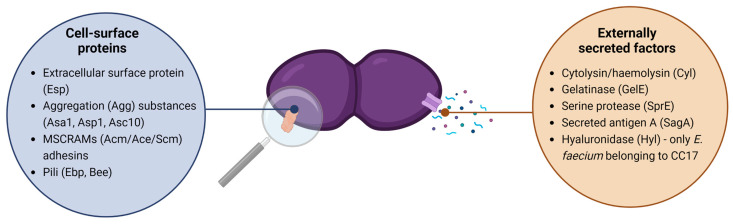
*Enterococcus* spp. most relevant virulence factors. Created with BioRender.com. Adapted from references [1,30].

**Figure 2 antibiotics-12-01140-f002:**
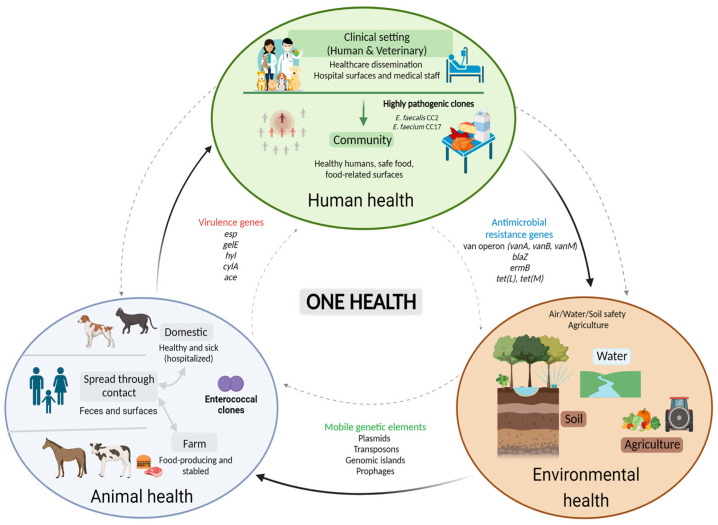
One Health sectors (human, animal, and environmental health). Created with BioRender.com.

**Table 1 antibiotics-12-01140-t001:** Main antimicrobials to which the different species of enterococci are resistant (intrinsic and acquired resistance), the mechanisms of resistance, and associated genetic determinants. Adapted from references [1,39].

Antibiotic Class	Cellular Target	Species (Mostly Present)	Type of Resistance	Mechanism of Resistance	Associated Antimicrobial Resistance Genes
β-lactams	Peptidoglycan synthesis	All enterococci	Intrinsic	Low affinity of penicillin-binding proteins (PBP)	*pbp5*/*pbp4*
		*E. faecium*, *E. hirae*, *E. faecalis*	Acquired (ampicillin high-level)	Overproduction or PBP alterations lead to lower affinity; β-lactamase	-
Aminoglycosides	Protein synthesis (30 s)	All enterococci	Intrinsic (low-level)	Poor antibiotic uptake	-
		*E. faecium*	Intrinsic (moderate-level)	Modification of the antibiotic molecule	*aac*
		*E. faecium*	Intrinsic	Target modification with rRNA methyltransferase	*efmM*
		*E. faecalis*, *E. faecium*, *E. gallinarum*, *E. casseliflavus*	Acquired (high-level)	Modification of the antibiotic molecule	*aph*, *ant*, *aac-aph*
		*E. faecium*, *E. faecalis*	Acquired	Target modification (point mutations)	*-*
Ansamycins	DNA replication	*E. faecium*, *E. faecalis*	Acquired	Mutations in the gene that encodes the β-subunit of RNA polymerase	*rpoB*
Glycopeptides	Cell wall synthesis	*E. gallinarum*, *E. casseliflavus*	Intrinsic (low-level)	Production of D-Ala-D-Lac/D-Ala-D-Ser terminus peptidoglycan precursors	*van* operons (*vanA*–*vanM*)
		*E. faecium*, *E. faecalis*	Acquired (high-level)	Precursor modification	
Fluoroquinolones	DNA replication	*E. faecium*, *E. faecalis*	Acquired	Target-site modification (gene mutation in DNA gyrase and topoisomerase IV)	*gyrA*, *pacC*
Tetracyclines	Protein synthesis (30 s)	*E. faecium*, *E. faecalis*	Intrinsic/acquired	Target-site modification	*poxtA*
			-	Antibiotic efflux	*tet (K)*, *tet (L)*
			-	Target-site protection	*tet (M)*, *tet (O)*, *tet (S)*
Macrolides	Protein synthesis (50 s)	Most enterococci	Acquired	Ribosomal methylation	*ermA*, *ermB*, *ermC*, *ermS*, *ermV*
Oxazolidinones	Protein synthesis (23 s)	*E. faecium*, *E. faecalis*	Intrinsic/acquired	Target-site protection	*poxtA*, *optrA*
		*E. faecium*	Acquired	Target-site modification (point mutations)	*-*
Phenicols (Chloramphenicol)	Protein synthesis	*E. faecium*, *E. faecalis*	Acquired	CAT-encoding enzymes	*catA*, *catB* (mostly)
StreptoGramins	Protein synthesis (early/late stage)	*E. faecalis*, *E. avium*, *E. gallinarum*, *E. casseliflavus*	Intrinsic	Antibiotic efflux	-

**Table 2 antibiotics-12-01140-t002:** Most relevant healthcare-associated enterococcal STs. Information was gathered considering enterococcal species allocation, source, geographic location, virulence, and antimicrobial resistance genes.

Species	Source	Location (Date)	Clade	Clonal Lineages		Virulence Genes	Antimicrobial Resistance Genes	Ref.
				MLST Sequence Type	Clonal Complex			
*E. faecalis*	Septicemias and endocarditis	Worldwide	NA	ST2	CC2	*gelE*	*blaZ*,* aac6′-aph2”*, *vanB*	[61,62,63]
				ST6		*esp*, *gelE*	*blaZ*,* aac6′-aph2”*, *ermB*,* vanA*,* vanB*	[61,63]
	Healthcare-associated infections	Worldwide (first in Argentina the USA)	NA	ST9	CC9	*esp*, *gelE*	*blaZ*, *aac6′-aph2”*, *ermB*	[29,61,64]
				ST106		*gelE*	*aac6′-aph2”*	[61]
	Bacteremia	Poland	NA	ST28	CC87	*asaI*, *ace esp*, *gelEcyl*, *hyl*	*aac6′-aph2”*, *vanA*	[61,62,64]
*E. faecium*	Healthcare-associated infections	Worldwide (first in the USA)	Clade A	ST17	CC17	*acm*, *esp*, *hyl*, *fms*	*vanA*, *vanB*, *dfrG*, *tetM*, *msrC*, *ermB*, *aac6*′-*aph2*″	[37,65,66,67,68,69]
				ST18		*acm*, *esp*, *hyl*	*-*	[65,66,67,68,69]
				ST78		*-*	*-*	[65,68]
				ST203		*acm*, *esp*	*vanB*	[65,66,67,68,70]
				ST796		*acm*, *esp*	*vanA*, *vanB*, *dfrG*, *tetM*, *msrC*, *ermB*, *aac6*′-*aph2*″	[65,66,67,68,70]
				ST1421		* - *	*vanA*	[71]

NA—Non-applicable. Information on virulence or antimicrobial resistance genes, associated with each ST, was recovered from the cited reference source, which does not imply that the indicated STs may not harbor other genetic traits.

**Table 3 antibiotics-12-01140-t003:** Most relevant enterococcal STs present in animals, food, community, and environment. Information was gathered considering enterococcal species allocation, source, geographic location, virulence, and antimicrobial resistance genes.

Species	Clonal Lineages		Virulence Genes	Antimicrobial Resistance Genes	Source	Location	Ref.
	Clonal Complex	MLST Sequence Type					
*E. faecalis*	CC2	ST40	*GelE*	*tetM*	Cheese	Poland	[98]
		ST6	ND	*vanA*, *tetM*, *aac6′-Ie-aph2″-Ia*	Water in swine facility	Portugal	[99]
	CC21	ST21	ND	*tetM*, *tetL*, *aac6′-Ie-aph2″-Ia*	Air in facility; liquid manure		
		ST202	*gelE*, *efaA*, *ace*	*aac6’–aph2”* *-Ia*	Poultry meat	Korea	[100]
	CC82	ST170	*gdh*, *gyd*, *pstS*, *gki*, *aroE*, *xpt*, *yqiL*	ND	Duck’s cecum	China	[101]
	CC93	ST93					
	CC192	ST192					
	CC314	ST314					
	CC593	ST593					
	CC903	ST903					
	CC476	ST116			Duck’s skin		
	-	ST16, ST21, ST26, ST84, ST138/501, ST207, ST209, ST277, ST326, ST672, ST674, ST715	*tuf*, *hyl*, *ebpA*, *bopD*, *fss1*	*ermB*	WWTP	Canada	[102]
		ST29	ND	*dfrE*, *emeA*, *efrA*, *efrB*, *lsaA*, *vanA*	Cattle feces	Switzerland	[103]
	-	ST242	*efaA*, *ace*, *ebpA*, *ebpB*, *epbC*, *gelE*, *fsrB*	*-*	Bovine feces	Canada	[104]
	-	ST271	ND	*lsaA*, *tetM*, *tetL*, *dfrE*, *emeA*	Unamended soil	South Africa	[105]
	-	ST1004, ST1006	ND	*dfrE*, *lsaA*, *emeA*	Litter-amended soil		
*E. faecium*	CC5	ST5	ND	*vanA*, *tetM*, *ermB*	Swine feces in slaughterhouse	Denmark	[106]
		ST150	ND	*tetM*, *ermB*	Adult swine feces	Portugal	[99]
		ST185	ND	*vanA*, *tetM*, *tetL*, *ermB*	Soil, solid manure, and adult swine diarrheal feces		
			ND	*vanA*, *tetM*, *ermB*	Swine feces in slaughterhouse	Denmark	[106]
		ST66	*fms5*, *fms17*, *fms21*	*tetM*, *tetL*	WWTP	Poland	[107]
	CC9	ST433	ND	*tetM*, *tetL*	Air from swine facility	Portugal	[99]
		ST437	ND	*tetM*, *tetL*, *ermB*	Adult swine feed	Portugal	[99]
		ST29, ST57	*aptA*, *ddl*, *gdh*, *purK*, *gyd*, *pstS*, *adk*	ND	Poultry	Malaysia	[108]
	CC17	ST10	ND	*aac6′-Ii*, *ant6-Ia*, *aph3′- III*, *ermB*, *lnuG*, *eatA*, *tetM*, *tetL*, *dfrG*, *efmA*	Litter-amended soil	South Africa	[105]
			*efaA*, *ccf/6*	ND	Poultry	Poland	[109]
		ST17	*ptsD*, *sgrA*, *IS16*, *orf1481*, *esp*	ND	Dog food	Portugal	[110]
		ST18	*tuf*, *aga*, *efaA*, *sgrA*, *uppS*, *lisR*, *acm*, *esp*, *scm*, *bsh*, *tip/ropA*, *bopD*, *eno*, *rfbA-1*	ND	WWTP	Canada	[102]
		ST203	*aptA*, *ddl*, *gdh*, *purK*, *gyd*, *pstS*, *adk*	*-*	Poultry	Malaysia	[108]
		ST78	ND	*vanA*, *ermB*, *ant6-Ia*, *aac6′-Ie-aph2″-Ia*, *aph3′-IIIa*	Rabbit meat	Spain	[111]
		ST78	*fms20*, *fms14*, *ebpA*, *fms16*	ND	WWTP	Czech Republic	[112]
		ST132	ND	*vanA*, *aac6′-Ie-aph2″-Ia*	Water in swine facilities	Portugal	[99]
		ST431	ND	*tetM*, *tetL*, *ermB*	Swine facility dust		
		ST386	*esp/intA*, *fms5*, *fms17*, *fms19*, *fms21*	*ant6’-la*	WWTP	Poland	[107]
	CC18	ST273	*fms20*, *fms14*, *ebpA*, *fms16*	ND	WWTP	Czech Republic	[112]
	CC22	ST21	*fms17*, *fms19*, *fms21*	*ant6’-la*, *tetM)*, *tetL*	WWTP	Poland	[107]
		ST32	ND	*tetM*, *tetL*, *tetS*	Antiseptic, drinking water in a swine facility	Portugal	[99]
		ST55	*aptA*, *ddl*, *gdh*, *purK*, *gyd*, *pstS*, *adk*	ND	Poultry and swine	Malaysia	[108]
	CC94	ST40	*tuf*, *aga*, *efaA*, *sgrA*, *upp*, *lisR*, *acm*, *esp*, *scm*, *bsh*, *tip/ropA*, *bopD*, *eno*, *rfbA-1*	ND	WWTP	Canada	[102]
		ST361	*fms5*, *fms17*, *fms19*, *fms21*	*ant6’-la*, *tetL*	WWTP	Poland	[107]
		ST1754	ND	*aac6′-Ii*, *msrC*, *eatA*	Poultry litter	South Africa	[105]
	-	ST1700	ND	*aac6′-Ii*, *ant6-Ia*, *ant9- Ia*, *ermB*, *lnuB*, *lsaE*, *msrC*, *cat*, *tetL*, *tetM*, *eatA*			
	-	ST1752, ST1756	ND	*aac6′-Ii*, *ermB*, *msrC*, *tetL*, *tetM*, *eatA*, *efmA*			
	-	ST1753, ST1755		*aac6′-Ii*, *ermB*, *msr C*, *tetL*, *tetM*, *eatA*, *efmA*	Litter-amended soil		
	-	ST214, ST955	*efaA*, *acm*	*ermB*, *msrC*, *aac6′-Ii*, *tetL*, *tetM*, *tetO*	Bovine feces	Canada	[104]
	-	ST133	ND	*aac60-Ii*, *eatAv*, *cadA*, *cadC*, *copZ*, *czrA*, *merA*, *merR*, *tetW/N/W*, *vanA*, *zosA*	Swine feces	Switzerland	[103]
	-		ND	*aac6′-Ii*, *eatAv*, *efrA*, *msrC*, *tetM*, *vanA*	Environmental water	Switzerland	[113]
	-	ST425	ND	*vanA*, *erm (B)*, *tet (M)*	Poultry meat	Spain	[111]
	-	ST13	ND	*aac60-Ii*, *aadK*, *eatAv*, *vanA*	Poultry feces	Switzerland	[103]
	CC117	ST117	*efaA*, *sagA*, *malR*, *swpA*, *swpB*, *swpC*	*ant6-la*, *ant1*, *aph*, *lnuB*, *isaE*, *tetL*, *satA*, *erm_1*, *erm_2*, *aad6-la*	Fermented dry sausage	Italy	[114]

NA—not applicable; ND—not determined. WWTP—wastewater treatment plant.

## Data Availability

Not applicable.

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
