# Peer review of "Dissemination of Enterococcal Genetic Lineages: A One Health Perspective"

_antibiotics, 2023, doi:10.3390/antibiotics12071140_

Round 1
Reviewer 1 Report
Enterococci are lactic acid bacteria, possessing a flexible nature that allows them to colonize various environments and hosts such as water, soil, plants, intestinal tract of warm blooded animals and man-made products, including fermented foods and dairy products. Enterococci are comprising both pathogenic and commensal microorganisms. Under certain circumstances enterococci can cause infections such as bacteraemia, peritonitis, endocarditis etc. The survival of enterococci in the hospital environment can be attributed to multidrug resistance and pathogenic traits. E. faecalis and E. faecium are the Enterococcus species most commonly associated with infection and vancomycin resistance. Enterococci are now ranked as a nosocomial pathogen, especially E. faecalis, but although certain antibiotic-resistant, infectious strains of enterococci, including E. faecium, have been identified in hospital patients. In recent years enterococci have passed the barrier of prevalence in nature and have become one of the leading causes of nosocomial infections.
Some members are used as probiotics and in production of feed additives to improve animal growth, so foods of animal origin are often contaminated with enterococci. That are likely to contribute resistance genes, virulence factors in humans and their use must be consider because of their status between safe and potentially harmful origin. In this regard, recent studies in molecular epidemiology based on molecular fingerprinting, multi-locus sequence typing, phenotypic studies and whole-genome analyses have provided further evidences for their nature and also provide of evidence that enterococci antimicrobial resistance genes appear to spread between enterococci from different reservoirs. Subsequent emergence of infections in humans caused by resistant bacteria that originate from the animal reservoir is of great concern. On the other hand Enterococcus spp. are part of the gastrointestinal tract microbiota in healthy humans and may be present in significant numbers. Species of the genus Enterococcus are also part of the microbiota of the human breast milk. All these properties have determine the importance to investigate their virulence and dispersion capacity. Vancomycin-resistant enterococci (VRE) have spread rapidity and today are steadily increasing worldwide. VRE infections are difficult to treat, appearing usually as long-lasting hospital outbreaks. Investigating the origin and dissemination routes are of the most importance to evaluate appropriate measures to reduce the rates of VRE and virulence factors such as aggregation substance, adhesins, hemolysin, hyaluronidase, and gelatinase.
This review clarifies the clinical features of enterococci up to date and provides important insights intoenterococcal pathogenicity - virulence factors and antimicrobial resistance. As antibiotic resistance increases, concurrently with the decreasing development of new therapeutics, it is essential that focus and investigation into these subject continue. The authors describe the most important features mainly of E. faecalis and E. faecium and their grouped into distinct genomic clades based on various environmental sources. Also there are some comments on suggesting an interconnection between hospital strains and household pets. This is a new route for enterococcal dissemination. The review combine the main information over the last years and demonstrated the important role of investigating antibiotic resistance, virulence patterns within clinical, farm and environmental samples and evaluated their spread and transmission between humans and animals.
In general the review summarized the resent studies on diversity of enterococcal genetic lineages and highlighted the need for effective genomic studies on enterococcal pathogenic elements across the One Health sectors.
Author Response
Thank you very much for the careful revision of our manuscript, higlighted by the accurate summary provided. We are really glad that you found it interesting, updated and accurate. Best regards.
Reviewer 2 Report
General Comments:
The presented manuscript is a review about Enterococci in a One Health context. I have a few reservations about the study design please see major comments. Overall the manuscript could be an added value for general scientific community. But I see the need to clarify details and the overall structure and focus of the manuscript. Chapter 4 is written very well and I recommend to re-write the manuscript, focusing on chapter 4.2 which has the most recent information. Despite these concerns, I generally recommend the manuscript for publication in MDPI Antibiotics, after major revision.
Major Comments:
In general: I would recommend to replace the clonal complexes (CCs) in the manuscript with the appropriate sequence types (STs). Nowadays most scientists use STs, and regarding NGS, or cgMLST-based complex types (CTs). Further, several STs that cluster together in a CC show complete different epidemiology and should not be put together.
Line 30: To add a few more interesting and important characteristics: growth/survival at high temperatures as 60°C (survival in non-pasteurized food) and persistence of desiccation (survival in environments and abiotic surfaces) (https://pubmed.ncbi.nlm.nih.gov/28502769/)
Line 37: Enterococci are also known as contamination source of food; missing information about enterococci as probiotics
Line 50: citation of more recent publications?
Line 59: I am missing here a sentence declaring/explaining the intention of the review. What shall be the focus and e.g. which other reviews could offer information on topics not discussed in this work.
Line 65-67: this is a wrong statement and clearly stated differently in the cited article by Cattoir. Please check the details, the statement is just valid for hospital-adapted lineages of E. faecium. Other publications highlight the importance of CRISPR-Cas in antibiotic resistant enterococci (https://pubmed.ncbi.nlm.nih.gov/35340673/)
Line 68-72: this explanation is no added value for the review article, in opinion
Line 100: what about fitness factors? It seems a bit “old-school” to only consider virulence factors and antibiotic resistance genes, especially since the authors aim to discuss the success of clonal lineages in One Health. It is known, that also fitness factors as adaptive physiology, persistence under stress (heat, cold, desiccation), survival on abiotic surfaces, formation of biofilms, quorum-sensing are very important for succeeding, especially in high-pressure selective environments as hospitals, livestock, food-production.
Chapter 3.1: the authors should consider shortening this entire section. It is well written, but the impact, and the connection of this chapter with the entire manuscript scope (genetic lineages in One health) is not given.
Chapter 3.2: also this part should be shorter and re-written, it´s missing a focus on genetic lineages or One Health.
Chapter 4: this part is really well written and more focused on the main topic of the review article. I strongly recommend shortening the parts before and focus on everything starting with chapter 4. But for almost the entire chapter: please write more details about the time. Statements as “over the past few decades, an increasing number of infections have been reported worldwide, due to the adaptation of specific clones to healthcare settings, mainly belonging to clonal complex 17” (line 273-275) are very vaguely formulated. I believe this might be problematic, since over the last decades, the entire epidemiology of genetic lineages changed a lot and there are big differences in different countries. One example is Germany where since about 10 years different lineages with VanB are circulation, while in neighboring countries VanA-associated lineages are dominating. The authors mention this beginning in Line 283, but the mentioned/cited data is outdated (most recent publication 2018); for example in Germany the ST192 was replaced by ST117 (see https://pubmed.ncbi.nlm.nih.gov/33189998/). This continues the next pages, referencing e.g. the situation in Australia in 2011. The text is not structured in a logical way; it “jumps” between E. faecalis, E. faecium, between epidemiology in different countries and genetic characteristics of different lineages. This could be sorted in a logical order.
Chapter 4.2: this chapter shows more recent data and is the first one really focusing on the main topic of the manuscript. Table 2 is very important, I really enjoyed seeing this data. I would recommend to consider the following study to this overview (especially figure S2): (https://journals.asm.org/doi/full/10.1128/jcm.01686-18), since the authors compared One Health E. faecalis isolates (animal, food, human, wastewater) and showed the high genetic diversity and the correlation of ST6 with human-continuum (including wastewater).
Line 450-452: please add following publication and may be discuss its results with the other studies: https://pubmed.ncbi.nlm.nih.gov/30401778/
Line 465: Further, for the revision I recommend to add the following (2 weeks old) paper: https://www.frontiersin.org/articles/10.3389/fsufs.2023.1179078/full which is perfectly fitting the scope of the manuscript and underlines the findings by Elghaieb et al.
Minor Comments:
Line 56: E. casseliflavus
Line 61: change “reduced” to “conserved”? Since the core genomes are still variable and enabling a multitude of metabolic pathways in parallel
Line 62: accessory genome E. faecalis about 32 % (https://journals.asm.org/doi/full/10.1128/jcm.01686-18)
English language is appropriate for publication.
Author Response
First of all, the authors would like to acknowledge the careful attention given to the submitted document. The comments and suggestions provided will undoubtedly contribute to the improvement of the manuscript. We sincerely hope that the modified version corresponds to the reviewer’s high expectations, and state that we are willing to perform further alterations, if considered fundamental to achieve a high-quality review, as intended by our research team.
Major Comments:
In general: I would recommend to replace the clonal complexes (CCs) in the manuscript with the appropriate sequence types (STs). Nowadays most scientists use STs, and regarding NGS, or cgMLST-based complex types (CTs). Further, several STs that cluster together in a CC show complete different epidemiology and should not be put together.
ANSWER: Thank you very much for your observation. Although the molecular epidemiology of each ST is different, many authors continue to attribute STs to clonal complexes, i.e., it is still a widespread concept in the scientific community, especially regarding CCs with higher clinical relevance and worldwide dissemination (like CC2, CC9 and CC17). However, we are aware that most scientists are nowadays starting to refer to STs (with a variety of CTs) rather than focusing on CCs. We acknowledge that joining clinical STs by CCs might not be the best current approach given the vastness of STs, with distinct epidemiology, being allocated to the most relevant CCs. Accordingly, we have inverted the CCs and STs columns in Table 2, to better highlight the most relevant STs. However, we maintained the CCs information, since we consider it to be relevant for a review article. In fact, when compiling data regarding the other One Health sectors/isolate sources, most literature mentions CCs (and sometimes only CCs). Hence, since specific STs are not always provided by the reference sources used, we consider that coupling STs with (relevant) CCs can provide valuable information regarding the dissemination along the One Health triad, which is the focus of this review. Thus, in Table 3 we originally grouped the STs by clonal complexes (CCs) to easily demonstrate the dissemination of STs belonging to particular CCs in a variety of environments/sectors. As we consider Table 3 to be more effective using this structure, we maintained the original focus/organization, but we remain open to suggestions to improve it.
Line 30: To add a few more interesting and important characteristics: growth/survival at high temperatures as 60°C (survival in non-pasteurized food) and persistence of desiccation (survival in environments and abiotic surfaces) (https://pubmed.ncbi.nlm.nih.gov/28502769/)
ANSWER: Thank you for the suggestion. We included some enterococcal fitness features (section 3.1), focusing on stress resistance and survival under harsh conditions. However, we did not find any information regarding enterococcal growth at 60 °C in the suggested reference (nor in other research articles), only survival at 60 °C when exposed to sublethal conditions, and cross-tolerance to a wide range of stresses including heat (62 °C), as reviewed by Gaca and Lemos (2019). If the reviewer considers that the aforementioned information needs to be included, please help us find the correct reference source. Thank you in advance!
Line 37: Enterococci are also known as contamination source of food; missing information about enterococci as probiotics.
ANSWER: Thank you for your relevant observation. We believe we had mentioned enterococci as a contamination source in food (please check line 37). However, as suggested, we added information regarding their importance as probiotics and as starter/non-starter cultures.
“These attributes being crucial for successful use in food fermentation processes, either as starter or non-starter bacteria, and inclusion in probiotic formulations [5].”
Line 50: citation of more recent publications?
ANSWER: Although the cited references demonstrated the impact of vancomycin-resistant E. faecium in healthcare settings, we acknowledge that more update information is necessary. We found a recent study with the percentages of incidence of VREfm clinical isolates on several continents, and included this information in the revised version.
Line 59: I am missing here a sentence declaring/explaining the intention of the review. What shall be the focus and e.g. which other reviews could offer information on topics not discussed in this work.
ANSWER: Although this is a narrative review and not a systematic review (please see https://www.scielo.br/j/ape/a/z7zZ4Z4GwYV6FR7S9FHTByr/?format=pdf&lang=en), we added a paragraph declaring the aim of this review and an explanation of the methodology applied.
“The aim of this literature review is to describe enterococci most relevant features, focusing on the overall dissemination of virulence and antimicrobial resistance genetic traits across the One Health sectors. Literature search was mainly conducted on PubMed and ScienceDirect databases (including cross-references). Only the most recent reports (major focus on the last five years) addressing the dissemination of genetic determinants between sectors of the One Health triad were selected (further clarified in section 4).”
Line 65-67: this is a wrong statement and clearly stated differently in the cited article by Cattoir. Please check the details, the statement is just valid for hospital-adapted lineages of E. faecium. Other publications highlight the importance of CRISPR-Cas in antibiotic resistant enterococci (https://pubmed.ncbi.nlm.nih.gov/35340673/)
ANSWER: Thank you for highlighting this issue. We have corrected the statement by specifying that it applies only to clinical E. faecium isolates, and added information regarding the correlation between the CRISPR-Cas system and acquired antimicrobial resistance.
“The genome plasticity of clinical E. faecium can be explained by the lack of genome defense mechanisms limiting HGT, such as CRISPR-Cas and restriction-modification systems [17]. Accordingly, an inverse correlation between the presence of CRISPR-Cas locus and acquired antimicrobial resistance in E. faecalis was first demonstrated in 2010 [21]. Recently, the lack of CRISPR-Cas genes was also associated with higher antimicrobial resistance rates and multidrug resistance in E. faecalis and E. faecium originating from hospital wastewater [22]”.
Line 68-72: this explanation is no added value for the review article, in opinion.
ANSWER: We have included this explanation to introduce the nomenclature (CCs and STs) used further on. Nevertheless, we mention this again in chapter 4.1 (lines 281 and 282 of the current version), thus we have removed it from lines 68-72, as suggested.
Line 100: what about fitness factors? It seems a bit “old-school” to only consider virulence factors and antibiotic resistance genes, especially since the authors aim to discuss the success of clonal lineages in One Health. It is known, that also fitness factors as adaptive physiology, persistence under stress (heat, cold, desiccation), survival on abiotic surfaces, formation of biofilms, quorum-sensing are very important for succeeding, especially in high-pressure selective environments as hospitals, livestock, food-production.
ANSWER: Thank you for highlighting the importance of mentioning fitness/adaptative features, given the aim of this review. Considering the relevance of your suggestion, we changed the title of section 3 to “Enterococcal fitness factors, virulence traits and antimicrobial resistance” and introduced topic 3.1, presenting some fitness factors that allow enterococci to survive and thrive in harsh environments. Although the overall structure of 3.2. and 3.3. sections were maintained, the information provided was summarized, as suggested.
Chapter 3.1: the authors should consider shortening this entire section. It is well written, but the impact, and the connection of this chapter with the entire manuscript scope (genetic lineages in One health) is not given.
Chapter 3.2: also this part should be shorter and re-written, it´s missing a focus on genetic lineages or One Health.
ANSWER: Thank you for the suggestions. For these sections, we aimed to introduce the main virulence factors and antimicrobial resistance patterns in enterococci (due to the scope of the journal and this special issue). We believe that this preliminary summary is fundamental to allow an easier interpretation of the information included in section 4, the main goal of this review. However, considering the reviewer’s suggestion, we tried to shorten/summarize the details provided, focusing on the most relevant. We hope that all the above can explain why we didn’t focus on genetic lineages or One Health - because we aim to gather that information only on section 4. Nevertheless, we highlighted the relevance of each of these sections considering the main focus of the article.
Chapter 4: this part is really well written and more focused on the main topic of the review article. I strongly recommend shortening the parts before and focus on everything starting with chapter 4. But for almost the entire chapter Answer: please write more details about the time. Statements as “over the past few decades, an increasing number of infections have been reported worldwide, due to the adaptation of specific clones to healthcare settings, mainly belonging to clonal complex 17” (line 273-275) are very vaguely formulated. I believe this might be problematic, since over the last decades, the entire epidemiology of genetic lineages changed a lot and there are big differences in different countries. One example is Germany where since about 10 years different lineages with VanB are circulation, while in neighboring countries VanA-associated lineages are dominating. The authors mention this beginning in Line 283, but the mentioned/cited data is outdated (most recent publication 2018); for example in Germany the ST192 was replaced by ST117 (see https://pubmed.ncbi.nlm.nih.gov/33189998/). This continues the next pages, referencing e.g. the situation in Australia in 2011. The text is not structured in a logical way; it “jumps” between E. faecalis, E. faecium, between epidemiology in different countries and genetic characteristics of different lineages. This could be sorted in a logical order.
ANSWER: Thank you for your suggestions and corrections. We have now included information regarding the time, when specified in the studies used as literature source. As we understand your concerns, the focus of our article is not epidemiological, but rather a comprehensive overview of the genetic lineages disseminated along the various One health sectors, highlighting the characteristic virulence and resistance determinants of certain STs, when the information is available in the literature sources. To prevent a too extensive length of section 4, we tried to find the best strategy to give an overview on the dissemination of genetic traits, focusing not only on the main CCs associated with healthcare environments, but also the dominant STs. In addition, despite the vast information on the topics covered, the molecular characterization of STs at the level of virulence and resistance genes is not equally explored in the various studies found in the literature search. Initially, we structured the text to introduce Table 2 with some background information on the subject. Moreover, we tried to restructure section 4.1 completely, to make it more uniform and logical to the reader. The jumps you mentioned were sometimes due to comparisons or features that genetic lineages, CCs, and epidemiology in different countries have in common, that we considered important to mention. In conclusion, we took under consideration your comments and suggestions and reorganized section 4.1. It is fundamental to refer that only the added/changed sentences were highlighted in yellow to be easily identified, but please consider that the entire structure of the 4.1. chapter was altered. We sincerely hope that this novel version meets your expectations.
Chapter 4.2: this chapter shows more recent data and is the first one really focusing on the main topic of the manuscript. Table 2 is very important, I really enjoyed seeing this data. I would recommend to consider the following study to this overview (especially figure S2): (https://journals.asm.org/doi/full/10.1128/jcm.01686-18), since the authors compared One Health E. faecalis isolates (animal, food, human, wastewater) and showed the high genetic diversity and the correlation of ST6 with human-continuum (including wastewater).
ANSWER: Thank you very much for your appreciation and suggestion. We have included the information provided by the suggested study on the review.
Line 450-452: please add following publication and may be discuss its results with the other studies: https://pubmed.ncbi.nlm.nih.gov/30401778/
ANSWER: Thank you for your help on the update of information. We considered the suggested reference in the review.
Line 465: Further, for the revision I recommend to add the following (2 weeks old) papeAnswer: https://www.frontiersin.org/articles/10.3389/fsufs.2023.1179078/full which is perfectly fitting the scope of the manuscript and underlines the findings by Elghaieb et al.
ANSWER: Thank you once again for the suggestion. We added that study to our review.
Minor Comments:
Line 56: E. casseliflavus
ANSWER: Corrected. Thank you.
Line 61: change “reduced” to “conserved”? Since the core genomes are still variable and enabling a multitude of metabolic pathways in parallel
ANSWER: Corrected. Thank you for the suggestion.
Line 62: accessory genome E. faecalis about 32 % (https://journals.asm.org/doi/full/10.1128/jcm.01686-18)
ANSWER: We were unable to find that information in the indicated article. If you can point out another reference mentioning the percentage of the accessory genome in E. faecalis, we would be very grateful.
Reviewer 3 Report
It is recommended to review the writing of the entire document, because it presents a variability in the writing and especially when citing the works (authors) that support the information; considering the guidelines, indications or recommendations that the journal itself gives to the authors.
Only some conditions that are observed in the manuscript are considered (as evidence), but the entire document requires attention from the authors in order to unify the writing, style, writing and grammar.
many authors have been performing comparative genomic analyses of Enterococcus spp. as indicator bacteria, to study the transference of ARGs, virulence genes and MGEs determinants across the One Health continuum [a few examples 16,31,33,49,50].
being mostly present in clade A isolates (reviewed by [25]
Sanderson and colleagues [16] explored the
These authors found 31 E. faecium and 34 E.
Additionally, Noh and coworkers also
Freitas et al. (2022) demonstrated
[52,53]. It is important to take care of the style - in some sections the references are in a different font (type and size), also in bold or underlined
Author Response
ANSWER: Thank you for your observations and recommendations. We believe your suggestions were essential to unify the writing style and improve the general quality of the document. We have corrected the citing criteria along the manuscript, considering the guidelines provided by the journal. As so, every time we cite the authors directly (by name) and when referring to “many studies/many authors” (to avoid being repetitive in writing style), we have included the reference immediately after. We believe we follow the citation style and rules provided by Antibiotics. Different formatting styles probably happened between review/editing versions, but were also corrected along the document. Please check our corrections and improvements, and let us know if any other issues need our attention.
Reviewer 4 Report
The manuscript by Marques et al. is well written easily understandable and provides a thorough overview of the most representative studies that describe the current state of the knowledge on the enterococcal pathogenic traits in context of the One Health approach.
The figures and tables are very helpful, and the text does a good job for provide a general overview about the diversity of enterococcal genetic lineages.
Nevertheless, below you have some tips to even improve it:
The biggest problem of this manuscript is the fact that in its present form seems to be a large list of information, but without a discussion able to point up significative findings. The material and methods section are missing together with the description of the data mining process, the eligibility criteria and rejection criteria of the cited works, number of chosen and rejected works. keywords used in the data mining process and in the different databases. The articles have been selected with a solid and clear methodology? If yes, what?
A paragraph with a complete and clear description of the bibliographic research is mandatory for a review article in order to increase its value!
I would like to see a revised version that includes a materials and methods section!
Author Response
ANSWER: Thank you for your careful review of our manuscript and for your considerate appreciation. We consider this to be a narrative review and not a systematic review (please consult the definitions at https://www.scielo.br/j/ape/a/z7zZ4Z4GwYV6FR7S9FHTByr/?format=pdf&lang=en). For this reason, a materials and methods section is not typically included in this type of review, nor do we consider it to be a value to the manuscript. Nevertheless, we agree that it is mandatory to add information on literature search. Thus, we added a paragraph declaring the intention of the review and an explanation of the methodology applied.
“The aim of this literature review is to describe enterococci most relevant features, focusing on the overall dissemination of virulence and antimicrobial resistance genetic traits across the One Health sectors. Literature search was mainly conducted on PubMed and ScienceDirect databases (including cross-references). Only the most recent reports (major focus on the last five years) addressing the dissemination of genetic determinants between sectors of the One Health triad were selected (further clarified in section 4).”
In conclusion, the authors would like to acknowledge the careful attention given to the submitted document. The comments and suggestions provided undoubtedly contributed to the improvement of the manuscript. We sincerely hope that the modified version corresponds to the reviewer’s high expectations, and state that we are willing to perform further alterations, if considered fundamental to achieve a high-quality review, as intended by our research team.
Round 2
Reviewer 4 Report
The authors partially addresses to the raised concerns by the reviewer in the first round, but their smart answers sound convincing! Congratulation!